# Looking at Alzheimer’s Disease Pathogenesis from the Nuclear Side

**DOI:** 10.3390/biom11091261

**Published:** 2021-08-24

**Authors:** Laura D’Andrea, Ramona Stringhi, Monica Di Luca, Elena Marcello

**Affiliations:** Department of Pharmacological and Biomolecular Sciences, Università degli Studi di Milano, 20133 Milan, Italy; laura.dandrea@unimi.it (L.D.); ramona.stringhi@unimi.it (R.S.); monica.diluca@unimi.it (M.D.L.)

**Keywords:** Alzheimer’s disease, transcription, amyloid-β, AICD, tau, *APOE ε4*, nucleus

## Abstract

Alzheimer’s disease (AD) is a neurodegenerative disorder representing the most common form of dementia. It is biologically characterized by the deposition of extracellular amyloid-β (Aβ) senile plaques and intracellular neurofibrillary tangles, constituted by hyperphosphorylated tau protein. The key protein in AD pathogenesis is the amyloid precursor protein (APP), which is cleaved by secretases to produce several metabolites, including Aβ and APP intracellular domain (AICD). The greatest genetic risk factor associated with AD is represented by the *Apolipoprotein E ε4* (*APOE ε4*) allele. Importantly, all of the above-mentioned molecules that are strictly related to AD pathogenesis have also been described as playing roles in the cell nucleus. Accordingly, evidence suggests that nuclear functions are compromised in AD. Furthermore, modulation of transcription maintains cellular homeostasis, and alterations in transcriptomic profiles have been found in neurodegenerative diseases. This report reviews recent advancements in the AD players-mediated gene expression. Aβ, tau, AICD, and *APOE ε4* localize in the nucleus and regulate the transcription of several genes, part of which is involved in AD pathogenesis, thus suggesting that targeting nuclear functions might provide new therapeutic tools for the disease.

## 1. Introduction

Alzheimer’s disease (AD) is the main cause of dementia and is becoming one of the most burdening diseases of the 21 century. In 2018, a dementia prevalence of about 50 million people worldwide has been estimated, with this number projected to triple by 2050 [1]. The incidence of AD rises exponentially with age; the majority of cases occur after age 65, constituting late-onset AD, while cases occurring earlier than age 65 represent less than 5% of all cases and are termed early-onset AD. Clinical symptoms include gradual loss of cognitive, affective, and behavioral functions, leading to increased impairment in activities of daily life [2].

The progression of AD is called the AD continuum and includes three broad phases: preclinical AD, mild cognitive impairment (MCI) due to AD, and dementia due to AD [3,4]. The AD phase is further classified into the stages of mild, moderate, and severe, which reflect the degree to which symptoms interfere with the ability to carry out everyday activities. The length of each phase in the continuum is variable and influenced by age, genetics, biological sex, and other factors [5]. Although the current diagnosis of AD relies on a combination of neuropsychological evaluations, biomarker measurements, and brain imaging, the diagnosis of AD is defined by the presence of amyloid β (Aβ) and phosphorylated tau. Indeed, the disease is mainly characterized by the accumulation of extracellular Aβ in senile plaques and intracellular neurofibrillary tangles (NFTs) constituted by hyperphosphorylated tau protein [6].

Regarding AD pathogenesis, the amyloid hypothesis has become the dominant model and is guiding the development of potential treatments. According to this hypothesis, an imbalance between production and clearance of Aβ is an early and initiating factor that leads to an increase in Aβ levels, and thus to its aggregation in oligomers and later in plaques. Such events trigger synaptic failure and inflammatory responses together with microglial and astrocytic activation, altered neuronal ionic homeostasis, oxidative injury, tau hyperphosphorylation, NFTs formation, and eventually neuronal loss and dementia onset [7].

Aβ, first isolated from the meningeal vessels of AD patients [8], is a small peptide of 40–42 amino acids generated via proteolytic cleavages of the amyloid precursor protein (APP). APP is a transmembrane protein; the enzymes responsible for APP cleavage have been identified in the 1990s [9]. The coordinated activity of the β- and γ-secretases liberates the Aβ peptide from APP [10]. β-secretase cleaves at the N-terminus of the Aβ domain, releasing a soluble fragment (sAPPβ) into the extracellular space and leaving a C-terminal fragment of 99 amino acids (CTF99) inserted into the membrane [9]. Several studies have identified a membrane-bound aspartyl protease with structural similarities to the pepsin family, named β-site APP cleaving enzyme-1 (BACE1), as the enzyme responsible for the β-cleavage of APP [11,12,13]. CTF99 is then cleaved by γ-secretase at the C-terminus of the Aβ sequence to liberate the Aβ peptide and the cytoplasmic APP intracellular domain (AICD). A heterotetrameric complex mediates γ-secretase activity and is constituted by Presenilin (PS), nicastrin (NCT), anterior pharynx defective (APH)-1a or APH-1b, and the PS enhancer (PEN)-2 [14]. PS contains the two critical Asp residues within the transmembrane domains 6 and 7 that are part of the aspartyl protease catalytic domain of γ-secretase [15]. The γ-secretase complex can produce Aβ peptides in lengths of 38, 40, and 42 amino acids that are released from the complex [16].

An alternative and nonamyloidogenic physiological pathway implicates the activity of α-secretase, which prevents Aβ generation and leads to the release of the neuroprotective soluble APP ectodomain (sAPPα) and the formation of an APP C-terminal fragment of 83 amino acids (CTF83) [17]. The metalloprotease A Disintegrin and Metalloproteinase 10 (ADAM10) is an enzyme capable of cleaving APP within an Aβ sequence and which exerts the main α-secretase activity in neuronal cells [18,19]. The γ-secretase then cleaves the CTF83 fragment partially inserted in the membrane to produce p3 peptide and the AICD.

The Aβ monomers generated by APP cleavage can aggregate to form oligomers, protofibrils, and fibrils. The aggregation rate of Aβ peptides is strongly correlated with the ratio of Aβ42/Aβ40 peptides in cellular and animal models [20,21]. Accumulation and further aggregation of protofibrils and fibrils lead to the formation of insoluble amyloid plaques, the main histopathological signature of AD [7]. However, in the past few decades, several studies have demonstrated that soluble forms of Aβ, rather than the large insoluble fibrils in plaques, are toxic to synapses [22]. Both synthetic and naturally-secreted forms of Aβ oligomers inhibit long-term potentiation (LTP) both ex vivo [23] and in vivo [24] and lead to frank loss of spines [25]. In addition, it has been shown that Aβ oligomers may drive cognitive deficits in animal models of AD [26,27] and potentially also in AD patients [28,29].

As the other classical hallmark of AD, NFTs represent neuronal cytoplasmic aggregates of tau protein that form paired helical filaments [30]. The abnormal phosphorylation of tau is the first step in the formation of these aggregates. Afterward, tau dissociates from axonal microtubules and aberrantly accumulates in the somatic cytoplasm and dendrites. Somatodendritic accumulations of phosphorylated tau form before fibrils and are called pretangles [31]. NFTs consisting of paired helical filaments of tau develop later and correlate with neuronal degeneration. In the human brain, silver-stainable NFTs develop in the transentorhinal cortex as well as in a few subcortical brain stem nuclei [32] and then spread into the entorhinal and other brain regions in a distinct hierarchical sequence that is different from that of Aβ-plaques [33].

Remarkably, in support of the amyloid hypothesis, mutations causing inherited forms of AD lead to increased production of Aβ and the development of AD, while mutations in the *Microtubule-Associated Protein* (*MAPT*) gene encoding tau do not cause familiar AD, although they can lead to other neurodegenerative tauopathies which often include Parkinsonian symptoms [34]. Indeed, more than 200 mutations in the genes coding for *PS* and about 20 in *APP* genes cause familiar cases of AD in an autosomal dominant manner. Such mutations are localized in Aβ sequences, close to the secretase cleavage sites, and affect the enzyme-substrate complex, leading to an increase in Aβ or to an increase in the Aβ42/Aβ40 ratio [14]. These mutations in *APP* or *PS* have also been exploited to generate animal models of the disease [35,36]. These inherited forms of AD represent approximately 1%–2% of AD cases, can present with very early ages of onset and a more rapid rate of progression, and are sometimes associated with other neurologic symptoms seen less frequently in sporadic AD [37].

In the case of late-onset AD, the greatest risk factors are advanced age and the presence of the *APOE ε4* allele [38]. Interestingly, the relatively rare *APOE ε2* allele remains by far the strongest genetic protective factor against sporadic AD, highlighting the importance of *APOE*’s role in AD pathogenesis. Relative to carrying the most common *APOE ε3* homozygous genotype, carrying at least one *APOE ε4* allele increases the risk of developing AD approximately 3.7 times, given *APOE ε4* homozygous increases that risk up to 12 times. Conversely, possessing a single *APOE ε2* allele reduces the risk by approximately 40%, and *APOE ε2* homozygous reduces the risk even further [39,40]. The post-mortem neuropathological factors correlated with the *APOE* genotype are a higher Aβ plaque burden and more severe cerebral amyloid angiopathy in *APOE ε4* carriers [41], also confirmed by Aβ Positron Emission Tomography imaging across preclinical and clinical AD stages [42].

*APOE ε4* affects Aβ pathology because it promotes the seeding of Aβ peptide into Aβ oligomers, protofibrils, and fibrils [43,44] while also inhibiting Aβ clearance from the brain, prolonging its half-life in the interstitial fluid [45,46] and inhibiting its enzymatic degradation [47]. In addition, in the last few years, the comprehension of the role of *APOE* in AD pathogenesis has expanded beyond Aβ peptide-centric mechanisms to NFTs degeneration, microglia, astrocyte responses, and blood–brain barrier disruption [48].

In light of these considerations, AD pathogenesis can be considered multifaceted and, therefore, difficult to pinpoint because it is the result of a complex interplay of crossing pathways. In particular, it has been shown that several elements implicated in AD pathogenesis can translocate to the nucleus and regulate transcription, thereby creating a network of pathways leading to AD dementia. This review summarizes the state-of-the-art understanding of and research into the AD players working as regulators of transcription and their potential contribution to AD pathogenesis. The description of such mechanisms is relevant for the development of therapeutic strategies aimed at modulating transcription in AD.

## 2. APP Metabolites and Their Role as Transcription Regulators

As described in the introduction, APP undergoes a consecutive shedding and intramembrane proteolysis mediated by secretases that can be summarized under the term “regulated intramembrane proteolysis” [49]. This is a cellular process that is frequently implicated in important signaling pathways [50,51], also involves the nucleus. Among APP metabolites, the intracellular domain AICD, which is released into the cytosol, has been extensively studied regarding its function in nuclear signaling [52,53]. However, in the last decade, the nuclear function of Aβ42 has also been investigated, also showing DNA binding properties of this peptide.

### 2.1. APP Intracellular Domain as a Potential Transcription Factor

A growing number of studies have reported the nuclear role of AICD in transcriptional control. Cytosolic AICD is subjected to rapid degradation [54,55], which leads to a short half-life [56]. Nevertheless, binding with the adaptor protein Fe65 stabilizes it and promotes its nuclear translocation [52,56,57]. Fe65, also known as the Amyloid Beta Precursor Protein Binding Family B Member 1 (APBB1), is a multifunctional protein belonging to the Fe65 family with the related proteins Fe65L1 and Fe65L2. It was recognized as an APP [58] and AICD [52,59] binding partner in yeast two-hybrid screenings. Fe65 cytosolic/nuclear localization is modulated by APP binding [60] and phosphorylation [61], as well as by its own phosphorylation [62]. When associated with APP, Fe65 is anchored to the membrane [60]; the γ-secretase-mediated APP processing releases the AICD-Fe65 complex and allows for its nuclear translocation [57,63]. Beyond its role in the nucleus, Fe65 has been reported to be involved in key cellular processes, such as actin dynamics [64] and calcium homeostasis [65], both impaired in AD.

AICD-Fe65 nuclear functions are modulated by the assembly of ternary complexes with different factors, such as Tip60 or CP2/LSF/LBP1. One well-characterized AICD-Fe65 coactivator is the acetyl transferase Tip60 [52,53,66]. In the nucleus, this complex can be detected as a dot-like structure [53,67]. AICD/Fe65/Tip60 complex is involved in the regulation of actin cytoskeleton, controlling the expression of *Stathmin1* (*STMN1*), a regulator of microtubules dynamics [67]. Nevertheless, the role of AICD in the modulation of *STMN1* transcription requires further elucidation.

It has further been shown that AICD/Fe65/Tip60 complex acts on other promoters. For example, it can activate the transcription of *KAI1* gene, replacing the N-CoR corepressor complex [66]. In the context of AICD-mediated gene expression control, it is important to underline that AICD regulates, both in vitro and in vivo, the expression of *Neprilysin* (*NEP*) [68,69,70], a protease involved in Aβ degradation. Thus, the AICD-mediated *NEP* regulation suggests a feedback mechanism in the control of APP metabolism. In HEK293 cells, the ternary complex AICD/Fe65/Tip60 works as an activator of *NEP* promoter, triggering a consequent increment in the protein levels [69]. Furthermore, chromatin immunoprecipitation (ChIP) experiments have confirmed that AICD binds *NEP* promoter (contributing to its activation) in NB7 cells and rat primary cortical neurons through an epigenetic mechanism involving a competition between AICD and Histone deacetylases [68]. Although several studies have demonstrated a contribution of AICD in *NEP* transcription, this regulation mechanism remains controversial [71].

AICD/Fe65 complex also associates with CP2/LSF/LBP1 family, inducing the expression of glycogen synthase kinase 3β (GSK3β) and leading to neurotoxicity as a result of tau phosphorylation increment and β-catenin levels reduction. This subsequently induces apoptosis [72]. Accordingly, other studies in neuroblastoma SHSY5Y cells have proposed the nuclear AICD as an encoder of pro-apoptotic signals through the association with the p53 factor [73].

Protein localization and function are finely tuned by post-translational modifications. Phosphorylation and SUMOylation modulate AICD nuclear activity [74,75]. Thr66 phosphorylation stabilizes AICD and is crucial for its localization and transcriptional activity [74,76], promoting the formation of a ternary complex with Fe65 and CP2 transcription factors and activating *GSK3β*, thus leading to neurotoxicity [74]. In accordance with these processes, other studies have shown that AICD controls *GSK3β* expression and activity in different cell types and primary neurons [53,72,77,78]. It has been recently reported that the SUMOylation of AICD Lys43 increases the association with Fe65 and nuclear localization [75]. Furthermore, SUMOylated AICD competes with the histone deacetylase 1 (HDAC1) for interaction with cyclic AMP (cAMP) response element-binding protein (CREB) and p65, controlling *NEP* and *TTR* expression [75].

Although Fe65 is a well-characterized AICD coactivator, a study conducted by Zhou and colleagues showed that AICD can also control transcription independently of Fe65 [79]. Indeed, in neuronal cells, Wnt signaling triggers the accumulation of AICD in the nucleus, where it associates with β-Catenin. This facilitates its activity as a transcriptional trans-activator [79]. Wnt and β-catenin constitute a signaling pathway fundamental for cellular homeostasis. When the Wnt signal is off, APC, GSK3β, and CK1 complex promote β-catenin ubiquitination and degradation. Conversely, when Wnt is activated, β-catenin translocates to the nucleus, where it controls gene expression. It is important to note that in brains affected by AD, Wnt signaling is compromised [80]. AICD and Wnt/β-catenin pathways are connected by a negative feedback loop: Wnt signaling promotes AICD-β-catenin interaction in the nucleus [79], and in turn, AICD promotes the formation of GSK3β-Axin complex and β-catenin ubiquitination and degradation [81]. Finally, AICD-mediated Wnt signaling regulation is involved in neuronal development, promoting the differentiation of neuronal cells and blocking their proliferation [81].

Other studies have shown that AICD can negatively regulate neurogenesis in vivo in APP KO mice and AICD mouse models [82,83,84], and in vitro can inhibit the differentiation of human neural stem cells (hNSC) [85,86]. A recent study showed that AICD cooperates with Forkhead box O3a (FOXO3a) in the control of neurogenesis in neural stem cells (NSCs) and neural progenitor cells (NPCs) derived from the hippocampus of an AICD mouse model [87]. A negative feedback regulation has been proposed between AICD and FOXO3a: AICD promotes FOXO3a transcriptional activity, which in turn negatively regulates AICD transcription. Accordingly, it has been shown that AICD drives FOXO3a transcriptional activity in primary cortical neurons [88].

Mitochondrial structure, function, and dynamics are affected in AD, and an emerging role of AICD has been proposed in AD mitochondrial dysfunction [89]. In collaboration with FOXO3a, AICD activity is implicated in mitochondrial functions; in particular, AICD controls the transcription of *Phosphatase and Tensin Homolog* (*PTEN*)-*Induced Kinase 1* (*Pink**-1*) in a FOXO3a-dependent manner [89]. Pink-1 has a cytosolic and mitochondrial localization, and its AICD-FOXO3a-mediated regulation has repercussions on mitochondrial functions and mitophagy [89]. Taken together, the above-mentioned studies suggest that, in the nucleus, AICD-mediated transcriptional modulation controls key molecular pathways that result in compromised processes leading to AD.

### 2.2. Amyloid-β Nuclear Localization and DNA Binding Properties

In the last decade, research has identified and examined the nuclear function of Aβ42. An important contribution to the study of Aβ as a putative transcriptional regulator arises from the in vitro identification of a specific Aβ binding domain on DNA [90]. An electrophoretic mobility shift assay (EMSA) identified the DNA guanine-rich sequence “KGGRKTGGGG” (where K is guanine or thymine, and R is a purine) as an Aβ-interacting consensus sequence. Such a sequence is recognized in the promoter of genes involved in AD pathogenesis, including *APP*, *BACE1*, and *APOE* [90]. Other works have confirmed that Aβ induces the expression of *BACE1*, suggesting a role in the regulation of its own metabolism [91].

In concert with these studies, ChIP experiments have shown that Aβ binds the *APP* promoter [92], and in SH-SY5Y neuroblastoma cells, Aβ exposure leads to an increase in *APP* transcription [93]. Conversely, in cortical primary neurons, Aβ exerts an opposite effect on *APP* mRNA levels, inducing suppression of its expression [91]. Moreover, such a discrepancy might be dependent on the dose and the time of exposure and on differences in the regulation mechanisms occurring in diverse kinds of cells.

Importantly, confocal imaging and electron transmission microscopy approaches have confirmed that Aβ42 localizes in the nuclei of neuroblastoma cells and APP/PS1 hippocampal neurons, supporting a nuclear role of Aβ42 [93]. A genome-wide DNA microarray conducted in neuroblastoma cells led to the identification of genes upregulated on Aβ exposure. Among them, gene encoding was identified for the *Insulin-Like Growth Factor Binding Protein 3* and *5*, whose levels increase even in the transgenic mouse expressing human APP (TgCRND8 mouse line) [94]. It is important to note that Aβ might act even as a transcriptional repressor, as reported for *Irp1* and *KAI1* promoters [29].

Notably, oxidative stress [92] and antibiotic treatment [95] trigger Aβ nuclear translocation, suggesting that Aβ-mediated changes in gene expression are required upon different stimuli in response to cellular stress. Although Aβ localizes in the nucleus [93] and has DNA binding properties [90,92], the mechanisms by which it enters the nucleus are not understood, and Aβ partners in the transcriptional control remain unidentified. Furthermore, whether the exogenous and endogenous intracellular Aβ might reach the nucleus through the same mechanisms and trigger the activation/repression of the same nuclear pathways has not yet been clarified. In this regard, the employment of patient-derived human induced pluripotent stem cells (iPSCs), a potent tool for modeling brain diseases, might provide an important contribution. Although different studies on AD took advantage of such cellular system, Aβ nuclear localization and function have not been explored in iPSCs yet. Moreover, iPSC-derived neurons might represent an important tool for understanding Aβ nuclear trafficking and its relevance in gene expression control. These observations leave many open questions regarding Aβ nuclear functions, but at same time present new perspectives for therapeutic interventions aimed at restoring physiological activity.

## 3. The Relevance of Nuclear Tau in Alzheimer’s Disease Pathogenesis

The tau protein was first described as an essential cytosolic factor involved in stabilizing microtubule assembly [96]. As reported above, tau aggregates are a common feature of several neurodegenerative disorders (termed tauopathies), including AD. Tau is expressed in the adult human brain in six different isoforms, which derive from the alternative splicing of exons 2, 3, and 10 of the *MAPT* gene, located on chromosome 17q21.1 [97,98]. The resulting proteins can have 0 (0N tau isoform), 1 (1N tau isoform), or 2 (2N tau isoform) inserts in the N-terminal domain, and 3 (3R tau isoform) or 4 (4R tau isoform) microtubule binding repeats in the C-terminal region [99]. Although the majority of current research on neuronal tau has focused on its role in the cytoplasmic compartment, a nuclear form of tau has also been characterized. In particular, it has been shown that each tau isoform has a preferred subcellular localization and thus a precise function [100]. In addition, the tau protein undergoes post-translational modifications and, therefore, it is important to understand whether these modifications can further influence its localization to the nucleus and its functional relevance.

Various effective studies on nuclear tau have been conducted, but several discrepancies have also emerged. In general, both phosphorylated and dephosphorylated isoforms have been detected in the nuclear compartment, which may vary depending on cell type and intranuclear localization. For example, the nuclear compartment of neuronal cells is characterized by the presence of dephosphorylated forms of tau [101,102,103,104,105]. Moreover, in human brains, dephosphorylated forms of tau have a specific localization in the nucleolus [102,106,107], while in murine neurons, it is diffusely expressed in the whole nuclear compartment [104]. Several in vitro studies have demonstrated the capability of tau to bind DNA [108,109]. These findings have also been confirmed in a cellular environment: 14% of total tau localizes in the chromatin fraction containing DNA, chromatin, and associated proteins [101].

Data obtained from cell cultures as well as mouse and human brains have established the involvement of nuclear tau in mechanisms of nucleoplasmic and ribosomal DNA (rDNA) protection against different cellular stressors, such as thermal denaturation [110] and hydroxyl free radical (–OH)-triggered double-strand DNA breakage [111,112]. Such stresses induce changes in tau phosphorylation states, causing the dephosphorylation of cytoplasmic tau and the consequent nuclear shuttling. The stressor dismission coincides with a decrease in nuclear tau to basal levels, while the phosphorylated level of cytoplasmic form increases. These findings suggest that tau phosphorylation/dephosphorylation represents a major mechanism of regulation for the trafficking of tau from the nucleus to the cytoplasm [113].

Tau–DNA interaction is mediated through tau’s proline-rich domain (PRD) and microtubule binding domain (MBD) with the DNA minor groove in order to form protein–DNA complexes [114]. However, tau has an additional interaction with DNA—more precisely, it co-localizes with dimethylated lysine 9 of histone 3 (H3K9me2)-rich DNA sequences, creating further protein–DNA complexes with the AT-rich α-satellite DNA sequences, organized as constitutive heterochromatin [107]. In order to highlight the tau protein’s physiological role, tau has been overexpressed in *Drosophila*
*melanogaster*, and it has been shown that it relaxes heterochromatin [115]. On the other hand, experiments conducted on KO-tau mice have highlighted its function as an arranger of nucleolar and/or heterochromatinization of a portion of antisense noncoding RNAs [116]. In addition, the depletion of tau results in an increase in rDNA transcription with an associated decrease in heterochromatin and DNA methylation, suggesting that under physiological conditions, tau is involved in silencing the rDNA. How tau is able to affect chromatin conformation remains unclear. However, it has been described that tau associates with TIP5 in the heterochromatin, a key player in genome stability [117].

Notably, the widespread loss of heterochromatin organization, which could be rescued by overexpressing nuclear human tau protein in KO tau neurons, has also been observed in AD neurons that displayed pathological hyperphosphorylated tau [116]. Heterochromatin loss likely permits increased expression of nonprotein-coding RNA transcripts. Indeed, increased expression of a regulatory RNA has already been implicated in the pathogenesis of AD. In particular, a noncoding antisense transcript for *BACE1* (BACE1-AS) is augmented in human AD brains, triggering upregulation of *BACE1* mRNA and protein levels [118]. The loss of heterochromatin, and the consequent aberrant gene expression, are toxic effectors of tau-induced neurodegeneration, underlining the role of chromatin structure as a potential therapeutic target in AD [115].

A further epigenetic mechanism involved in the organization of the chromatin structure and reflecting changes in gene activity is H3K9 acetylation (H3K9ac). Klein and colleagues used H3K9 acetylation (H3K9ac) as a marker in an epigenome-wide association study to reveal a tau-driven disorganization of chromatin in AD affecting 5990 out of 26,384 H3K9ac domains [119]. Together, these findings suggest that AD involves a reconfiguration of the epigenome, and the identification of this process highlights potential epigenetic strategies for early-stage disease treatment [119,120].

Tau is also detectable in a specific region of acrocentric chromosomes of dividing cells, called the nuclear organizer region (NOR). This site is enriched in rRNA genes and represents the sub-nucleolar compartment where the formation of nucleolus takes place [121,122]. A post-translational modification of tau involving phosphorylation plays a key role in regulating its function and interaction. Indeed, NOR-associated tau is dephosphorylated, and its phosphorylation induces tau detachment from the DNA [114]. Since tau hyperphosphorylation is a hallmark of AD, this event can enhance tau–DNA dissociation. Notably, in AD patients, the dentate gyrus (DG) and the CA1 region of the hippocampus display an altered mRNA expression and altered protein levels in rRNA transcription key regulators, such as upstream binding factor (UBF), nucleolin (NLC), and nucleophosmin (NMP1). These factors are decreased in the AD hippocampi from the first stages of pathology, even before neuron loss [123]. These alterations may represent an early sign of AD, which precedes the reduction in dendritic arborization and the synaptic reduction in CA1 and DG neurons, finally resulting in hippocampal atrophy.

A recent study highlighted a further transcriptional pathway aberrantly changed in AD. Through examination of AD post-mortem brain tissues and *Drosophila*
*melanogaster* models, a direct correlation between the increased activation/mobilization of transposable elements (TEs) and the pathogenic activity of tau has been demonstrated [124]. This event represents a key driver of anomalous cell cycle activation in neurons and subsequent neuronal death. Among upregulated transposable elements in human tauopathy, the human endogenous retrovirus (HERV) family (including HERV-K) has emerged [125]. Interestingly, a causal association between HERV-K and neuronal dysfunction has previously been established, as the expression of HERV-K or the retroviral envelope protein that it encodes decrease synaptic activity in mice [126].

Recent research has further described a mechanism of transcription regulation governed by tau, lacking a direct interaction with DNA but depending on the modulation of nuclear calcium (Ca^2+^) levels. During synaptic plasticity events, an influx of Ca^2+^ occurs, resulting in the regulation of transcription through the activation of CREB. Using an animal model of tauopathy (tau^R406W^ Transgenic *Drosophila*) and iPSC-derived neurons from AD patients, Mahoney and colleagues highlighted an aberrant depletion of nuclear Ca^2+^. Synapse-to-nucleus communication is mainly governed by Ca^2+^ [127], and pathogenic tau directly contributes to CREB and calcium depletion. Indeed, CREB-regulated genes are significantly over-represented in tau^R406W^ transgenic *Drosophila*. In addition, pharmacological activation of big potassium (BK) channels is able to restore nuclear Ca^2+^ levels and suppress neurodegeneration in tau^R406W^ transgenic *Drosophila* [128]. Although it is not yet evident whether the regulation of such genes is a direct consequence of tau-induced CREB depletion, these findings provide a deeper understanding of the involvement of tau in AD pathogenesis.

## 4. *APOE ε4* as a Transcriptional Regulator

As described in the introduction, *APOE ε4* allele represents the strongest genetic risk factor for AD [38]. In the mammal brain, *APOE* proteins are mainly expressed in astrocytes and microglia; moreover, neurons may produce them under stress and in reaction to toxic events [129,130,131,132]. Interestingly, it has been shown that *APOE* proteins can translocate to the nucleus, where they are able to bind DNA and function as a transcription factor in human glioblastoma cells [133]. Furthermore, in neuroblastoma, *APOE ε4* negatively regulates the *Brain-Derived Neurotropic Factor* (*BDNF*) through a mechanism involving the enhancement of the nuclear translocation of histone deacetylases [134], suggesting that *APOE ε4* controls the expression of key genes involved in brain functioning.

In a recent work, the effect of *APOE* proteins was tested on human neuronal cultures depleted of glia to exclude the contribution of the secreted proteins to the *APOE*-mediated phenotype in neurons. Neurons were plated on MEFs and transfected with recombinant *APOE ε2*, *APOE ε3*, and *APOE ε4* [135]. From such experiments, an intriguing mechanism regarding *APOE*-mediated signaling has emerged. *APOE* is, in fact, able to activate a noncanonical MAP kinase signaling pathway. In particular, *APOE* binds the receptors on the cell surface, triggering the activation of map kinase kinase Dual Leucine Zipper Kinase (DLK), which phosphorylates the map kinase Mitogen-Activated Protein Kinase Kinase 7 (MKK7), leading to the phosphorylation of the MAP kinase Extracellular Signal-Related Kinase 1/2 (ERK1/2). Such a signaling cascade results in the stimulation of c-Fos phosphorylation and in the consequent AP-1-dependent *APP* transcription, which triggers an increment in Aβ production, with an efficacy increasing from *APOE ε2* to *APOE ε3* to *APOE ε4* [135]. Notably, other studies have shown that *APOE ε4* interacts with APP and controls its metabolism, inducing a decrease in sAPP production [136].

The advent of new technologies allowing for the generation of 2D and 3D cultures derived from iPSCs is a strong in vitro tool and was recently implied in combination with the gene editing technique CRISPR/Cas9 for the study of *APOE ε4* contribution to AD [137]. Although in this study, a direct *APOE ε4*-mediated regulation of gene expression was not assessed, transcriptomic analysis conducted in *APOE ε4*-expressing neurons showed that the genes that were primarily downregulated were associated with synaptic functions [137]. In addition, *APOE ε4* neurons show an increment in Aβ42 with respect to the isogenic line expressing *APOE ε3* [137], thus confirming the *APOE ε4* impact on AD-related dysfunctions. Furthermore, ChIP experiments followed by high-throughput DNA sequencing (ChIP-seq) led to the identification of *APOE ε3* or *APOE ε4* target genes [133]. A total of 1700 genes were found regulated by *APOE ε4* and not by *APOE ε3*, 76 of which were related to AD [133].

It is important to underline that *APOE* is involved in the transcriptional control of APP pathway players. The promoter of *Sirtuin1* (*SirT1*), a transcriptional activator of the α-secretase ADAM10 gene [138], has been identified among the promoters bound by *APOE*. In particular, *APOE ε4* acts as a transcriptional repressor of *SirT1* [133]. Accordingly, *APOE* ε4 expression leads to a reduction in *SirT1* levels in AD brains and primary neurons [136]. The role of *APOE ε4* as a negative transcriptional regulator was confirmed for at least three other genes, i.e., *MAP Kinase Activating Death Domain* (*MADD)*, *Activity Dependent Neuroprotective Protein (ADNP)*, and *COMM Domain 6* (*COMMD6)* [133]. Moreover, it is not possible to exclude an additional function as a transcriptional activator. These studies provide evidence that *APOE* proteins play a key role in the modulation of neuronal transcriptome; in particular, they indicate that *APOE ε4* controls (both directly and indirectly) the expression of genes implicated in AD, confirming its contribution to this pathology. As described above, the isogenic conversion of *APOE ε4* to *APOE* ε3 ameliorates AD-related phenotypes in AD iPSCs-derived brain cell types [137] and iPSCs-derived cerebral organoids [139]. Furthermore, *APOE ε4*, but not *APOE ε3*, induces changes in the expression of AD related genes [133]. Such observations suggest that the genetic conversion *APOE ε4* allele to *APOE ε3* isoform might represent a promising therapeutic tool. The CRISPR/Cas9 approach allows the genetic correction of DNA mutation and constitutes a new frontier for molecular biology research and gene therapy. Once ethical concerns are overcome, this technique might provide a powerful instrument for long-lasting genetic modifications aimed at restoring gene functions as in the case of *APOE ε4*-*APOE ε3* conversion.

## 5. Conclusions

A growing number of studies have focused on nuclear and transcriptional dysfunctions in AD [140,141,142]. In the era of omics, high-throughput approaches, and big data generation, many reports have highlighted changes in the transcriptomic profiles of AD for in vitro and in vivo models and patients [143,144,145,146,147,148]. In addition, gene association studies have identified several putative AD-associated risk factors genes [149]. Such studies shed light on the unexplored mechanisms involved in the pathogenesis of this neurodegenerative disorder. Moreover, other research has provided new insights into the nuclear function of the “canonical” AD-associated molecules, such as the APP metabolites Aβ and AICD, tau, and *APOE ε4* (Figure 1).

Aβ, AICD, and *APOE* are able to bind DNA [66,68,72,79,88,90,92,133,138] and act as transcriptional regulators. Importantly, they modulate the expression of key players of AD pathogenesis. *APOE* regulates *APP* [136] and the transcriptional activator of ADAM10 secretase, *SirT1* [133]. Aβ can bind *APP*, *BACE1*, and *APOE* promoters [90], while AICD regulates, both in vitro and in vivo, the expression of *Neprilysin* [68,69,70], a protease involved in Aβ clearance, just to cite the most relevant regulated genes (Figure 1 and Table 1). Such evidence suggests a feedback mechanism that must be finely tuned in physiological conditions; such equilibrium is disrupted in AD, in a vicious cycle that feeds itself.

Tau has also been described to play nuclear functions, and a nuclear form of tau has been identified. When tau protein is dephosphorylated it is involved in DNA protection against cellular stressors [111,112,113], as well as in the rearrangement of a heterochromatin portion enriched in noncoding RNAs [117] (Figure 1 and Table 1). Since AD pathogenesis is characterized by abnormal tau hyperphosphorylation, the loss of heterochromatin and the resultant anomalous gene expression could be the toxic effectors of tau-induced neurodegeneration. Effectively, altered levels of rRNA transcription key regulators are reported in post-mortem analysis of AD brains from the first stages of pathology [123]. Overall, these findings provide new insights into tau-induced alterations, suggesting potential strategies for early-stage disease treatment.

All the studies examined in this report underline the importance of proper nuclear signaling in the context of AD and indicate that correction of transcriptional defects might provide a novel therapeutic approach.

## Figures and Tables

**Figure 1 biomolecules-11-01261-f001:**
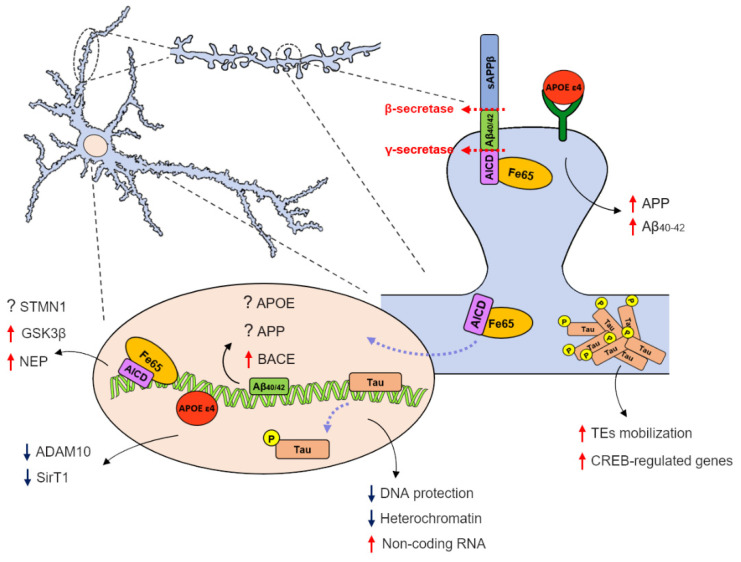
Schematic representation of how AD effectors contribute to neuronal dysfunction by modulating transcription. APP undergoes intramembrane proteolysis mediated by β- and γ-secretases to generate Aβ and AICD. The intracellular metabolite AICD binds Fe65, which promotes its nuclear translocation and the consequent interaction with the DNA. The AICD/Fe65/DNA complex controls the expression of *Stathmin1 (STMN1),* activates *Neprilysin* (*NEP*) promoter, and induces the expression of *Glycogen Synthase Kinase 3β* (*GSK3β*). The Aβ monomers, generated by APP cleavage, translocate to the nucleus and recognize a sequence in the promoter of genes involved in AD pathogenesis, including *APP*, *BACE1*, and *APOE*. Tau hyperphosphorylated aggregates result in an anomalous activation of transposable elements (TEs) and an aberrant CREB-regulated genes over-representation. When nuclear tau is dephosphorylated, it is involved in DNA protection against cellular stressors and the rearrangement of heterochromatin, enriched in noncoding RNAs. AD pathogenesis is characterized by abnormal tau hyperphosphorylation, causing tau detachment from DNA and the consequent loss of heterochromatin and anomalous noncoding RNAs expression. *APOE ε4* binding to its receptor activates a noncanonical signaling pathway which triggers the *APP* transcription and Aβ40/42 production. When *APOE ε4* interacts with DNA, it acts as a transcriptional repressor of *Sirtuin1* (*SirT1*), a transcriptional activator of the α-secretase *ADAM10* gene. (Red and blue arrows indicate the increase and the decrease in gene transcription respectively. Question mark (?) shows an unknown effect (activation or repression) of transcription).

**Table 1 biomolecules-11-01261-t001:** Nuclear functions and target genes of AD players.

Molecules	DNA Binding	Nuclear Function	Target Genes	References
AICD	yes	Transcription regulator	*STMN1* *KAI1* *NEP* *GSK3β* *Pink-1*	[67] [66] [68,69,70,75] [72,74] [89]
Aβ	yes	Transcription regulator	*APP* *BACE* *APOE* *IRP1* *KAI1*	[90,93] [90] [90] [29] [29]
tau	yes	DNA protection Chromatin remodeling Genome stability	Unknown genes	[111,112] [107,115] [117]
*APOEε4*	yes	Transcription regulator	APP SirT1 *MADD* *ADNP* *COMMD6* 1700 other genes	[135,136] [133,138] [133] [133] [133] [133]

## Data Availability

Not applicable.

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
