# Peer review of "Looking at Alzheimer’s Disease Pathogenesis from the Nuclear Side"

_biomolecules, 2021, doi:10.3390/biom11091261_

Round 1

Reviewer 1 Report

The article titled “Looking at Alzheimer´s Disease pathogenesis from the nuclear side” by D´Andrea et al. is an updated review on recent results that suggest that AD molecular players can mediate gene expression. Altough this a well-written article, it could benefit if the following ideas / concepts are taking into account:

1. Regarding Ab42 localization in the nucleus, has this been found in human iPSC-derived neurons? Please, discuss this point in the revised version of the manuscript.

  1. The authors quote reference 135 when discussing the effect of APOE proteins on human neuronal cultures. However, this APOE effect was published by the group of T.C Südhof in Cell (reference 139). Have the findings reported in this article been replicated by other research groups working on human iPSC-derived neurons?

  1. The Reference list should be revised throughout for correcting several mistakes, including the one mentioned above as well as references 52 and 54 (author names, titles and journal details are missing) and other potential mistakes.

Reviewer 2 Report

The review article is very interesting and the authors did explain the new term of nuclear side which is the main highlight of the review. The current review is organized very well. 

The Author did explain the amyloid hypothesis as a major contributor for AD. 

It would be interesting if authors can explain if the nuclear side also affects the different receptors function, especially the G-protein which plays an important role in AD. 

Could APOE ε4 be a drug target if so then the author needs to include a paragraph stating the designed experiment? 
